# Angular X-ray Cross-Correlation Analysis (AXCCA): Basic Concepts and Recent Applications to Soft Matter and Nanomaterials

**DOI:** 10.3390/ma12213464

**Published:** 2019-10-23

**Authors:** Ivan A. Zaluzhnyy, Ruslan P. Kurta, Marcus Scheele, Frank Schreiber, Boris I. Ostrovskii, Ivan A. Vartanyants

**Affiliations:** 1Department of Physics, University of California San Diego, La Jolla, San Diego, CA 92093, USA; ivan.zaluzhnyy@gmail.com; 2Deutsches Elektronen-Synchrotron DESY, Notkestrasse 85, D-22607 Hamburg, Germany; 3European XFEL, Holzkoppel 4, D-22869 Schenefeld, Germany; ruslan.kurta@xfel.eu; 4Institute of Physical and Theoretical Chemistry, University of Tübingen, Auf der Morgenstelle 18, 72076 Tübingen, Germany; marcus.scheele@uni-tuebingen.de; 5Center for Light-Matter Interaction, Sensors & Analytics LISA+, University of Tübingen, Auf der Morgenstelle 15, 72076 Tübingen, Germany; frank.schreiber@uni-tuebingen.de; 6Institute of Applied Physics, University of Tübingen, Auf der Morgenstelle 10, 72076 Tübingen, Germany; 7Federal Scientific Research Center “Crystallography and Photonics”, Russian Academy of Sciences, Leninskii Prospect 59, 119333 Moscow, Russia; ostrenator@gmail.com; 8Institute of Solid State Physics, Russian Academy of Sciences, Academician Ossipyan str., 2, 142432 Chernogolovka, Russia; 9National Research Nuclear University MEPhI (Moscow Engineering Physics Institute), Kashirskoe Shosse 31, 115409 Moscow, Russia

**Keywords:** angular X-ray cross-correlation analysis, orientational order, soft matter, hexactic phase in liquid crystals, colloidal crystals, organic-inorganic mesocrystals

## Abstract

Angular X-ray cross-correlation analysis (AXCCA) is a technique which allows quantitative measurement of the angular anisotropy of X-ray diffraction patterns and provides insights into the orientational order in the system under investigation. This method is based on the evaluation of the angular cross-correlation function of the scattered intensity distribution on a two-dimensional (2D) detector and further averaging over many diffraction patterns for enhancement of the anisotropic signal. Over the last decade, AXCCA was successfully used to study the anisotropy in various soft matter systems, such as solutions of anisotropic particles, liquid crystals, colloidal crystals, superlattices composed by nanoparticles, etc. This review provides an introduction to the technique and gives a survey of the recent experimental work in which AXCCA in combination with micro- or nanofocused X-ray microscopy was used to study the orientational order in various soft matter systems.

## 1. Introduction

Soft matter, which includes polymers, liquid crystals, colloidal systems, membranes, foams, and granular materials, covers a broad range of possible condensed matter structures between periodic crystals and completely disordered liquids. Usually these systems are characterized by relevant length scales above interatomic distances and weak interparticle interactions, which are typically of the order of thermal energies at room temperature [1,2]. As the result, soft matter is very sensitive even to small changes of external conditions, such as pressure, temperature, and electromagnetic fields [3].

The strong response to external stimuli and rich phase diagrams has given rise to numerous applications of soft matter materials in electronics, display devices, photonics, photovoltaics, coatings, and adhesives, to name a few. Another important property of many, but not all, soft matter systems is their ability to self-assemble, i.e., form complex ordered structures from basic units, such as nano- or colloidal particles [4]. Self-assembly allows one to fabricate new artificial materials by controlling macroscopic parameters, such as concentration and temperature, instead of manipulating each basic unit on a microscopic level. On the other hand, such self-assembled structures are prone to defects, which obviously have influence on the resulting functional properties of these materials.

Understanding the structure of various soft matter systems is an important yet challenging problem of modern condensed matter physics. First of all, soft matter typically exhibits more degrees of freedom compared to conventional inorganic crystals. This is related to the fact that the basic units of soft matter, for example, organic molecules or nanoparticles, are much larger than atoms, and exhibit angular degrees of freedom, so not only their position, but also orientation in space become important for describing the structure of these systems. It is interesting to note that these orientational degrees of freedom were speculated to be the reason for fundamental differences in the non-equilibrium structure formation between molecular and atomic systems [5]. Moreover, they are the reason for the very existence of liquid crystals (LCs) and a broader range of structures for multi-component molecular systems [6]. Second, many soft matter systems are known to display a structural hierarchy, for example, amphiphilic LC molecules may form bilayers or micelles, which in turn are organized in complex superstructures. Finally, due to the weak interaction between structural elements of soft matter systems and long relaxation times, different thermodynamically distinct structures may replace each other before the crystallization occurs.

X-ray scattering is a well-established tool used to investigate the structure of complex materials. One of the main advantages of X-rays compared to other probes is high penetration length, which allows to study the bulk structure of the materials, in contrast to surface electron scattering, for example. For decades, the use of X-ray beams provided the means to obtain information on the average structure of soft matter and other forms of condensed matter. Nowadays, X-ray beams from modern coherent synchrotron sources can be focused below one hundred nanometers, which allows one to study the local structure by measuring the X-ray diffraction from a very small illuminated sample volume. It opens a unique opportunity to study the local arrangement of the structural elements in soft matter systems. However, many challenges, such as low signal-to-noise ratio and radiation damage, require further development of techniques for experimental data analysis, which would allow one to extract as much information from the experimental data as possible.

In this review, we report on basic concepts and recent applications of angular X-ray cross-correlation analysis (AXCCA) [7], a novel method which is capable to provide insights into orientational order in the system and characterize it quantitatively. Initially, this method was proposed by Z. Kam [8,9] to determine the structure of bioparticles by analyzing angular correlations of the scattered intensity from a dilute solution of identical particles. In parallel, the first measurements of angular correlations in structural studies of colloidal systems were reported using light scattering in optical range [10,11]. In the last decade, it was realized that the analysis of X-ray angular correlations can provide otherwise hidden information on the structure of a dense systems, such as colloids [12,13], organic films [14,15], LCs [16,17], and mesocrystals [18,19].

The present paper elucidates recent progress of AXCCA applied to soft condensed matter systems and is organized as follows. In the second section, a brief description of different types of structural order is given with the focus on the bond-orientational order. The third section provides basic theoretical description of the quantities used in AXCCA. The fourth section covers the application of the AXCCA to selected systems, such as LCs in the vicinity of the smectic-hexatic phase transition, colloidal systems, organic-inorganic mesocrystals, and solvated metal complex molecules. The fifth section completes the paper with a summary and outlook of the possible further applications of AXCCA.

## 2. Systems with Angular Anisotropy

### 2.1. Classification of Structural Order

Let us consider a homogeneous system consisting of identical particles, such as atoms, molecules or colloidal particles. The structure of such a system can be described by the pair-correlation G(r) Equation (1):(1)G(r)= 〈ρ(r)ρ(0)〉
where ρ(r) is a probability to find a particle at the position r (single particle distribution function) and brackets 〈…〉 denote ensemble averaging. In the case of an ideal one-dimensional (1D) crystal lattice, which is characterized by a periodic arrangement of particles, the function G(r) will look as it is shown in Figure 1a. The sharp peaks at positions r=na, where n is an integer and a is the average distance between neighboring particles, indicate that there is a high probability to find the particles at these separations from the given one. If the magnitude of the peaks approaches a constant nonzero value for r→∞, this type of order is called long-range. In a generic X-ray diffraction experiment, one can measure the structure factor in Equation (2):(2)S(q)=∫G(r)exp(−iqr)dr,
which is the Fourier image of the pair-correlation function G(r) [1]. Thus, in diffraction patterns, the long-range order manifests itself by the presence of sharp Bragg reflections, as it is schematically shown in Figure 1b. Due to reduced dimensionality, thermally excited displacements and various defects, the peaks of G(r) can be broadened and their magnitude may decay with distance. A special case when this decay is algebraic, i.e., it can be described by a power law $\propto r^{-\eta }$, is called quasi-long-range order. In this case, the magnitude of the higher reflections of the structure factor S(q) decreases and the peaks become broader.

In the case of a simple liquid, the pair-correlation function G(r) behaves quite differently (see Figure 1c). Instead of sharp peaks at distinct separations, G(r) decays much faster, namely, as ∝exp(−rξ), where ξ is the correlation length [21]. Such an exponential decay of the correlation function is a distinctive feature of the short-range order (Figure 1c). Performing the Fourier transform, one can show that the structure factor S(q) in this case is characterized by a single broad Lorentzian-like scattering peak centered at ~2π/a (Figure 1d). The width of the peak is inversely proportional to the correlation length ξ, which allows one to study the decay of the correlation function by analyzing the shape and width of the X-ray scattering peak.

Clearly, the pair correlation function is only sensitive to the positional order, which makes this formalism suitable for systems composed of particles with a spherical shape. However, there are many examples of the assemblies of particles possessing an anisotropic shape, so the orientational degrees of freedom are coming into the play. This situation is typical for various phases of LCs with elongated or disk-like molecules as well as certain polymers [22]. In this case the distribution function ρ includes not only the position of particles but also their orientation, ρ=ρ(r,Ω), where Ω is the angular coordinate that describes the particle’s orientation [23,24].

In different systems the function ρ(r,Ω) can take different forms. For example, in the case of the nematic liquid crystal, in which centers of mass of elongated molecules are essentially randomly distributed but all molecules are oriented along the common direction ***n*** (Figure 2a). Therefore, the distribution function can be written as ρ(r,Ω)=ρf(cosθ), where ρ is a constant and the orientation distribution function f(cosθ) depends only on the polar angle between Ω and ***n***, i.e., cosθ=(Ωn). In this case the orientational order parameter in nematics can be defined by expanding f(cosθ) in series of orthogonal polynomials of cosθ, i.e., the Legendre polynomials $P_{l}\left(\cos\theta \right)$ [25].

An example of more ordered LC phase is a smectic-A phase of parallel molecular layers, in which elongated molecules in each layer are oriented on average along the layer normals (Figure 2b). In this phase the single particle distribution function ρ(r,Ω) is periodic along the direction z (perpendicular to the molecular layers), so ρ(r,Ω)=ρ(z,Ω), while across the layers there is a short-range positional order. There are many other soft matter systems where various types of order occur in different directions or at different scales. An example of such a system is the columnar phase of LCs, formed by flat-shaped discotic molecules [25,26] (Figure 2c). In this phase the flat molecules form columns along the *z*-direction with short-range positional order, i.e., the molecules are randomly distributed along the column. However, the top view of the columnar phase reveals that the projections of the columns on the plane perpendicular to the orientation of the columns form a periodic two-dimensional array. This corresponds to a quasi-long-range positional order with the density function of the form ρ(r,Ω)=ρ(x,y,Ω).

### 2.2. Bond-Orientational Order

The concept of the bond-orientational (BO) order has emerged as one of the breakthrough approaches in condensed matter physics. It appeared first regarding the problem of melting in 2D crystals, where the hexatic phase possessing the BO order was predicted as an intermediate state between a crystal and a liquid [27,28,29]. The BO order is determined by a specific angular arrangement of the particles and it can be seen as the presence of preferred angles between interparticle “bonds”, which are virtual lines connecting centers of mass of the neighboring particles [30,31,32]. Thus, the BO order differs from the orientational order of anisometric (non-spherical) particles discussed in the previous section, where the orientation of the particles can be described in terms of the single-particle distribution function ρ(r,Ω).

To illustrate the simplest case of the BO order, let us consider a dense 2D system of particles, as shown in Figure 3a. The local orientation of bonds at the spatial position r can be described by the angle θ(r) between the bond and some reference axis. If local *n*-fold rotational symmetry is present in the system, one can introduce the BO order parameters as Equation (3):(3)$$\Psi_{n}\left(r\right)=\langle e^{in\theta \left(r\right)}\rangle ,$$
where the brackets 〈…〉 indicate a coarse grain average [1,32]. In the case of the *n*-fold rotational symmetry, the average angle between the bonds is equal to 2π/n, so all exponential terms corresponding to different bonds in the coarse grain average will have the same phase and they will sum up constructively. In contrast, for a completely disordered system the angles between bonds are random, and the coarse grain average in Equation (3) will lead to a zero value of $\Psi_{n}\left(r\right)$. Thus, it is natural to combine the magnitude and the phase of the BO ordering field within the complex BO order parameter, see Equation (4):(4)$$\Psi_{n}\left(r\right)=\psi_{n}\left(r\right)e^{in\theta_{n}\left(r\right)}.$$
Here the value of $0\leq \psi_{n}\leq 1$ determines the magnitude of the BO order, while the phase $\theta_{n}\left(r\right)$ corresponds to the local orientation of the bond with respect to the reference axis.

We note that even in the simplest case of densely packed non-interacting spheres in 2D, the local value of $\Psi_{n}\left(r\right)$ is non-zero for n=6, due to the fact that the hexagonal arrangement of particles allows the highest packing density. This corresponds to the short-range hexatic order in isotropic liquids. However, even if the local arrangement of the particles resembles ideal hexagons, the orientational alignment of the hexagons rapidly falls off with the distance r. To quantify the persistence in orientation of local arrangement of particles across the entire system, the following orientational correlation function can be defined in Equation (5):(5)$$G_{n}\left(r\right)=\;\langle \Psi_{n}\left(r\right)\Psi_{n}\left(0\right)\rangle .$$

Similar to the pair correlation function defined in Equation (1), the exponential decay of $G_{n}\left(r\right)$ indicates short-range BO orientational order, while the algebraic decay of $G_{n}\left(r\right)$, or its approaching to constant nonzero value at r→∞ correspond to a quasi-long-range and long-range BO order, respectively.

Simple liquids are usually characterized by the short-range positional and orientational order with a typical correlation length of the order of few interparticle distances. Thus, in an X-ray diffraction experiment all orientations of bonds within the illuminated area are averaged, and the 2D diffraction pattern consists of a broad and uniform scattering ring as shown in Figure 3b. However, for systems with the long-range BO order, for example, the hexatic phase (Figure 3c), the modulation of the intensity along the azimuthal direction is visible [33,34,35,36,37,38] (Figure 3d).

The anisotropy of the diffraction pattern allows to experimentally quantify the BO order by applying the angular Fourier series decomposition of the 2D scattered intensity distribution. The scattered intensity in the detector plane with the polar coordinates (q,φ) (see Figure 3b,d) can be written as Equation (6):(6)$$I\left(q,\varphi \right)=I_{0}\left(q\right)\left(1+2\overset{\infty }{\underset{n=1}{\sum }}C_{n}\left(q\right)\cos\left(n\varphi +\varphi_{n}\left(q\right)\right)\right),$$
where $I_{0}\left(q\right)$ is the angular-averaged intensity at the momentum transfer q, $C_{n}$ are the normalized magnitudes of the cosine modulations along the ring, and $\varphi_{n}$ corresponds to the relative phases of these modulations. In an ideal crystal with *P*-fold rotational symmetry, the low order diffraction peaks at $q_{0}$ are separated by an angle 2π/P. In this case, the values of the magnitudes $C_{n}\left(q_{0}\right)$ should be equal to unity if *n* is divisible by *P*, and $C_{n}\left(q_{0}\right)=0$ for all other values of n. In the case of distortion of the orientational order in the crystal lattice, for example, due to the presence of domains with slight angular misorientation between them, the diffraction peaks will broaden in the angular direction. As a result, the values of $C_{n}\left(q_{0}\right)$ will decrease below unity for all n divisible by *P* and approach zero when separated diffraction peaks merge into a uniform scattering ring.

Since scattering at $q_{0}\sim 2\pi /a$ corresponds to the average interparticle distance a in real space, the coefficients $0\leq C_{n}\leq 1$ are sensitive to the orientation of bonds between neighboring particles separated by a. Thus, the set of coefficients $C_{n}$ can be considered as the BO order parameters determined in reciprocal space [17,39,40,41,42], in contrast to the complex-valued orientational field $\Psi_{n}\left(r\right)$ determined in real space.

### 2.3. Recent Developments in the Study of Soft Matter with X-rays

Our previous discussion of studies of the soft mater systems was focused on a situation when comparably large partially coherent X-ray beams were used. In this case, an illuminated part of the sample is typically much larger than the correlation length and in the scattering experiment we obtain information from the ensemble average at each illuminated spot. The development of the 3rd generation X-ray synchrotron sources opened new possibilities to study soft matter with coherent X-rays. The high degree of coherence of these sources enables interference between the X-rays scattered from different parts of the sample, which results in appearance of strong fluctuations of intensity in the diffraction patterns, i.e., speckles (see Figure 4). An apparently random ensemble of speckles encodes the instantaneous positions and orientations of the particles in the system. This leads to two different applications; if the system is non-static, then by studies of speckle fluctuations information on dynamics of the system may be obtained [43]. If the system is static, the complete information on the instantaneous structure of the system can be recovered by means of coherent diffraction imaging [44,45] or ptychography [46].

Another advantage of the 3rd generation X-ray synchrotron beams with high coherence is the ability to focus them to hundred nanometer sizes and below [47,48,49]. With such small focus sizes only a limited number of particles may be illuminated by a single X-ray spot. This will naturally lead to fluctuations of intensity when the X-ray spot will be moved from one part of the sample to another. In this case a raster scan of the sample with a focused X-ray beam may be performed to obtain structural information on ensemble average orientational order over a large area of the sample.

We should also note here that a few conditions should be fulfilled in order to observe speckles in addition to the high coherence of the X-ray beam. These include illumination times smaller than the characteristic time scale of the dynamics in the system; otherwise the collected diffraction patterns would be smeared due to time-averaging. The requirement of the short exposure times leads to another condition for the photon flux, which should be sufficiently high to produce enough scattering during the short X-ray pulse. Recently developed X-ray free electron lasers (XFELs) are capable to fully satisfy these requirements, which paves new ways for experimental studies of the structure of soft mater in general and the orientational order in particular.

## 3. Principles of AXCCA

Let us consider an X-ray diffraction experiment in transmission geometry as shown in Figure 4. To describe the angular anisotropy of the diffraction pattern one can consider the angular Fourier series of the scattered intensity I(q,φ) in Equation (7):(7)$$I\left(q,\varphi \right)=\overset{\infty }{\underset{n=-\infty }{\sum }}I_{n}\left(q\right)e^{in\varphi },$$
where the *n*-th Fourier component is Equation (8):(8)$$I_{n}\left(q\right)=\frac{1}{2\pi }\overset{\pi }{\underset{-\pi }{\int }}I\left(q,\varphi \right)e^{-in\varphi }d\varphi .$$

The non-zero modulus of the complex value $I_{n}\left(q\right)$ indicates the presence of the *n*-fold rotational symmetry in the diffraction pattern, and the phase of $I_{n}\left(q\right)$ corresponds to the angular positions of the maxima of this Fourier component with respect to the reference axis. Thus, the zero-order Fourier component $I_{0}\left(q\right)$ equals to the angular-averaged scattered intensity, and all other Fourier components with n≠0 define the angular modulations of the intensity along the ring of radius q on the detector.

AXCCA constitutes a general approach to describe and analyze the angular anisotropy of the diffraction patterns. This can be advantageous in the case of complex systems, where angular anisotropy is present on different length scales, or when the scattering signal needs to be averaged over many diffraction patterns for better statistics [7]. The basic element of AXCCA is the two-point angular cross-correlation function (CCF), which is defined as Equation (9):(9)$$\hat{C}\left(q_{1},q_{2},\Delta \right)=\overset{\pi }{\underset{-\pi }{\int }}I\left(q_{1},\varphi \right)I\left(q_{2},\varphi +\Delta \right)d\varphi ,$$
where −π≤Δ<π is the angular variable [50,51]. In the simplest case, when the orientational order in the system is characterized by a single length scale, one may consider $\hat{C}\left(q,\Delta \right)\equiv \hat{C}\left(q,q,\Delta \right)$. However, the CCF can also be evaluated for $q_{1}\neq q_{2}$ for the cases where angular correlations between two different length scales are studied. We should also note that AXCCA is not limited to the two-point CCF, and in some cases consideration of three- and higher order CCFs is required [38].

A natural way to investigate the symmetry properties of the CCF is to expand it into the angular Fourier series, see Equation (10):(10)$$\hat{C}\left(q_{1},q_{2},\Delta \right)=\overset{\infty }{\underset{n=-\infty }{\sum }}{\hat{C}}_{n}\left(q_{1},q_{2}\right)e^{in\Delta },$$
where the *n*-th Fourier component is Equation (11):(11)$${\hat{C}}_{n}\left(q_{1},q_{2}\right)=\frac{1}{2\pi }\overset{\pi }{\underset{-\pi }{\int }}\hat{C}\left(q_{1},q_{2},\Delta \right)e^{-in\Delta }d\Delta .$$

By definition, the CCF is a real-valued function, so its Fourier components satisfy the condition ${\hat{C}}_{-n}\left(q_{1},q_{2}\right)={\hat{C}}_{n}^{*}\left(q_{1},q_{2}\right)$, where the asterisk denotes the complex conjugate. In the simplest case of $q_{1}=q_{2}=q$, the definition of the CCF implies that it is an even function, i.e., $\hat{C}\left(q,\Delta \right)=\hat{C}\left(q,-\Delta \right)$, so only cosine terms are present in the Fourier series, see Equation (12):(12)$$\hat{C}\left(q,\Delta \right)={\hat{C}}_{0}\left(q\right)+2\overset{\infty }{\underset{n=1}{\sum }}{\hat{C}}_{n}\left(q\right)\cos\left(n\Delta \right).$$

The Fourier components ${\hat{C}}_{n}\left(q_{1},q_{2}\right)$ contain valuable information about the symmetry of the system and are directly related to the Fourier components of the scattered intensity $I_{n}\left(q\right).$ Applying the convolution theorem, one can show [50], see Equation (13):(13)$${\hat{C}}_{n}\left(q_{1},q_{2}\right)=I_{n}^{*}\left(q_{1}\right)I_{n}\left(q_{2}\right).$$
Direct comparison of Equations (6), (7), (13) reveals that for $q_{1}=q_{2}=q$ the Fourier coefficients ${\hat{C}}_{n}$ of the CCF are related to the BO order parameters as ${\hat{C}}_{n}\left(q\right)/{\hat{C}}_{0}\left(q\right)={\left|\frac{I_{n}\left(q\right)}{I_{0}\left(q\right)}\right|}^{2}=C_{n}^{2}\left(q\right)$, defined in Equation (6). Thus, the values of the BO order parameters can be directly evaluated by ACXXA.

One may consider ensemble-averaged values of the CCF or its Fourier components over many realizations of the system in Equation (14):(14)$$\begin{array}{c} \langle \hat{C}(q_{1},q_{2},\Delta )\rangle =\frac{1}{N}\overset{N}{\underset{j=1}{\sum }}{\hat{C}}^{j}\left(q_{1},q_{2},\Delta \right),\\ \langle {\hat{C}}_{n}(q_{1},q_{2})\rangle =\frac{1}{N}\overset{N}{\underset{j=1}{\sum }}{\hat{C}}_{n}^{j}\left(q_{1},q_{2}\right), \end{array}$$
where the index j≤N enumerates diffraction patterns corresponding to different realizations of the system. We should note that direct averaging of the 2D diffraction patterns or complex values $I_{n}\left(q\right)$ may lead to smearing of the signal, for example, in the case when the patterns differ from each other only by rotation around the central point q=0. At the same time, averaging of the CCFs would lead to an enhancement of the symmetry features of the diffraction patterns, while the experimental noise will be reduced. Angular anisotropies in X-ray or electron diffraction patterns may be analyzed as well by the multipole expansion of Legendre polynomials as suggested in reference [52].

To illustrate the type of problems which can be addressed by AXCCA, let us consider X-ray scattering from an ensemble of identical particles randomly oriented in space. First of all, one can recover the structure of an individual particle by analyzing the diffraction patterns from the ensemble of particles with a uniform distribution of orientations [53,54,55,56,57]. This problem becomes significantly simplified if one can neglect the interference between the diffracted X-rays from different particles, which is possible if the transverse coherence length of X-rays is smaller than the typical interparticle distance [51]. This corresponds to the case of a “dilute solution” in conventional scattering theory for liquids or small-angle X-ray scattering (SAXS), when the structure factor simply equals to unity.

The second type of problems is related to the quantitative description of the orientational correlation of particles. This includes both, the characterization of orientational distribution of particles in solutions [58,59] as well as the bond-orientational order in dense systems [12,16,17]. In the latter case, studies of the bond-orientational order necessarily include interference of the scattered X-rays between several particles [31,60], which can significantly complicate the analysis. A typical example of such a system is the hexatic phase in LCs, which will be considered in the following section.

## 4. Applications of AXCCA 

### 4.1. The Hexatic Phase in Liquid Crystals

In this section, we will focus on the development of the BO order in the vicinity of the smectic-hexatic phase transition. The smectic-A (Sm-A) phase in thermotropic LCs can be seen as a set of equidistant parallel molecular layers, in which elongated molecules are oriented perpendicular to the layer plane (Figure 2b). The top view of each layer reveals a liquid-like structure, which is characterized by the short-range positional and BO order (Figure 3a), as well as a zero value of the shear modulus.

In free-standing smectic films, the molecular layers are perfectly parallel to the surface of the film, which makes them an ideal object for X-ray scattering studies [20,25]. In the geometry depicted in Figure 5, the X-ray diffraction is sensitive to the in-plane structure of the molecular layers, which produces a broad scattering ring of radius $q_{0}\sim 4\pi /a\sqrt{3}$, where a is the average in-plane distance between the LC molecules [20,26]. Based on the phenomenological Ornstein-Zernike model [1,21,23], the radial cross-section of the scattering ring can be approximated by the Lorentzian function Equation (15):(15)$$I\left(q\right)\propto \frac{\gamma^{2}}{{\left(q-q_{0}\right)}^{2}+\gamma^{2}}$$
with the half width at half maximum (HWHM) γ=1/ξ, where ξ is the positional correlation length (compare with Figure 1c,d). Typical values of ξ vary in the range of 1 to 2 in-plane intermolecular distances [16,17,20], which is a clear indication of short-range positional order.

At lower temperatures, some LC compounds form the stacked hexatic phase (Hex-B), which differs from the Sm-A phase by the presence of the sixfold long-range in-layer BO order (Figure 3c). At the same time, the positional order remains short-range, however the positional correlation length increases compared to its value in the Sm-A phase. Due to the interaction between the molecules, the intermolecular bonds have approximately the same orientation in the neighboring layers, but the shear modulus between the layers remains zero. As a result, in reciprocal space the BO order in the Hex-B phase manifests itself by a sixfold modulation of the scattering intensity at $q_{0}$ along the azimuthal direction, which at lower temperatures leads to splitting of the scattering ring into six distinct peaks [20,61] (Figure 3d). It was shown that due to the interaction between the positional order and BO order thermal fluctuations, the radial cross-section through the centers of the hexatic peaks changes to the square root of the Lorentzian function close to the Sm-A–Hex-B phase transition [17,62], while far away from the transition temperature it is well described by the simple Lorentzian function (Equation (15)). The same interaction also slightly shifts the position of the scattering peaks along q over the value proportional to the mean squared modulus of the hexatic BO order parameter ${\langle |\Psi_{6}|}^{2}\rangle$ [62].

Some conclusions about the evolution of the BO order at the Sm-A–Hex-B phase transition may be drawn from considerations of the symmetry properties of these phases. The in-plane sixfold BO order in the Hex-B phase can be described in real space by the fundamental BO order field $\Psi_{6}\left(r\right)$ and its higher harmonics $\Psi_{6m}\left(r\right)$ (Equation (4)), where m>1 is an integer [63,64]. However, since the direct determination of the molecular orientations in LCs is not possible, in an experiment it is more common to use a set of the BO order parameters $C_{6m}$ defined in reciprocal space according to Equation (6) [39,63].

The two-component field $\Psi_{6m}\left(r\right)$ allows to describe the Sm-A–Hex-B phase transition within the XY universality class [1], which is characterized by a second-order phase transition in 3D (see the phase diagram in Figure 6). The transition of Sm-A into the hexagonal crystal phase Cr-B is always first-order in 3D due to the long-range positional order and the presence of a cubic term in the free-energy expansion in powers of the Fourier coefficients of the density [65]. The same cubic term is present for a Hex-B–Cr-B transition in 3D and gives rise to the first-order transition. The two lines of the first-order transitions intersect on the phase diagram at slightly different slopes, which also shifts the Sm-A–Hex-B phase transition to the first order [17,40]. This leads to the appearance of a tricritical point on the Sm-A–Hex-B phase transition line in the vicinity of the Sm-A–Hex-B–Cr-B triple point (Figure 6). At the tricritical point the Sm-A–Hex-B phase transition changes its order from first to second in the vicinity of the Sm-A–Hex-B–Cr-B triple point (Figure 6). As a result, in many LC compounds the Sm-A–Hex-B phase transition occurs in the vicinity of the tricritical point.

Assuming that the Sm-A–Hex-B phase transition lies in a crossover region between the mean field (tricritial behavior) and the 3D XY universality class, Aharony et al. [39,40] have shown that the BO parameters $C_{6m}$ for m=1,2,… satisfy the scaling relation, called the multicritical scaling theory (MCST) Equation (16):(16)$$C_{6m}={\left(C_{6}\right)}^{\sigma_{m}},$$
where the exponent $\sigma_{m}$ is Equation (17):(17)$$\sigma_{m}=m+x_{m}\cdot m\cdot \left(m-1\right).$$

The parameter $x_{m}$ can be represented as the Taylor series given in Equation (18):(18)$$x_{m}\approx \lambda -\mu m+\nu m^{2}.$$

Using the renormalization group approach, Aharony et al. [39,63] evaluated the value of the parameters of the MCST to be λ=0.3 and μ=0.008 in 3D hexatics. The theoretical estimation of the quadratic correction term ν has never been carried out because of the required excessive computational efforts. It was also shown that in the 2D hexatic phase $x_{m}=1$, which significantly simplifies the scaling relation (see Equations (16) and (17)) [66].

Experimental verification of the MCST predictions is complicated due to the necessity of the accurate determination of the BO order parameters $C_{6m}$. In the diffraction experiment on the 8OSI compound (racemic 4-(2′-methylbutyl)phenyl 4′-(octyloxy)-(1,1′)-biphenyl-4-carboxylate), by fitting the azimuthal dependence of the scattered intensity with Equation (6), it was possible to evaluate the values of $C_{6m}$ up to m=7, which allowed one to obtain λ=0.295 [39]. In Refs. [17,67] diffraction patterns from n-propyl-4′-n-decyloxybiphenyl-4-carboxylate (3(10)OBC), n-heptyl-4′-n-pentyloxybiphenyl-4-carboxylate (75OBC) and 1-(4′pirydyl)-3-(4-hexyloxyphyenyl- oamine)-prop-2-en-1-on (PIRO6) LC compounds were analyzed by AXCCA. The values of the BO order parameters $C_{6m}$ were directly evaluated as a square root of the corresponding normalized Fourier coefficients of the CCF $C_{6m}=\left|I_{6m}\left(q_{0}\right)\right|/\left|I_{0}\left(q_{0}\right)\right|=\sqrt{\left|{\hat{C}}_{6m}\left(q_{0}\right)/{\hat{C}}_{0}\left(q_{0}\right)\right|}$. Due to the spatial uniformity of the sixfold BO order in the Hex-B films over the range of about 100 μm in hexatic monodomains, the magnitude of the angular modulation along the scattering ring remained almost constant at different positions. This allowed averaging the CCFs over 100 diffraction patterns for the 3(10)OBC and PIRO6 compounds and over 25 patterns for the 75OBC compound. As a consequence of such evaluation of the BO order parameters, it was possible to determine up to $m_{\max}=25$ successive Fourier components for 3(10)OBC, and $m_{\max}=7$ for 75OBC, and $m_{\max}=6$ for PIRO6, respectively. The magnitudes of the angular Fourier components of intensity $\left|I_{n}\left(q_{0}\right)\right|$ in the Hex-B phase for different compounds are shown in Figure 7. The FCs of the order *n*, which are not divisible by six have vanishingly small values on the noise level. In contrast, the sixfold FCs are much larger, indicating the presence of sixfold BO order in the system. The decay of the magnitudes $\left|I_{6m}\left(q_{0}\right)\right|$ with m≥1 can be well fitted by the MCST law at any temperature point below the Sm-A–Hex-B transition for all three compounds (red lines in Figure 7).

Although the MCST predicts λ=0.3 for 3D Hex-B phase, the value of the parameter might change close to the Sm-A–Hex-B phase transition point. This was observed indeed for all three compounds [17,67], where λ was abruptly decreasing down to zero at the phase transition temperature (Figure 8a). However, away from the Sm-A–Hex-B phase transition temperature the experimentally obtained values λ=0.31±0.0015 (3(10)OBC), λ=0.27±0.0015 (75OBC), and λ=0.29±0.0015 (PIRO6) are in a perfect agreement with the MCST predictions [40]. Moreover, the unprecedentedly high number of the measured successive BO order parameters at low temperatures in 3(10)OBC compound, allowed the authors to evaluate a sufficient number of exponents $\sigma_{6m}$ (Figure 8b) in order to obtain the correction term μ=0.009±0.001 in Equation (18) and estimate the value of the quadratic correction $\nu \sim 10^{-4}$ [67].

While many of the known hexatic LC compounds exhibit a second-order or very weak first-order Sm-A–Hex-B phase transition [17,39,68,69], this transition in the freely suspended films of the n-pentyl-4′−n-pentanoyloxy-biphenyl-4-carboxylate (54COOBC) compound is clearly of the first order. In reference [70], an X-ray beam focused down to 2×2 μm^2^ was used to scan the 100×100 μm^2^ area of the 54COOBC film in transmission geometry (Figure 5). At each spatial position, AXCCA was utilized to evaluate the value of the fundamental BO order parameter $C_{6}=\sqrt{\left|{\hat{C}}_{6}\left(q_{0}\right)/{\hat{C}}_{0}\left(q_{0}\right)\right|}$. The values $C_{6}<0.1$ correspond to almost uniform scattering ring with negligible azimuthal modulations of intensity (Figure 9a). The corresponding position of the film was considered to be in the Sm-A phase. In the opposite case, points with $C_{6}>0.1$ (Figure 9b) were attributed to the Hex-B phase. The spatially resolved maps of the BO order parameter $C_{6}$, measured at different temperatures in the vicinity of the Sm-A–Hex-B phase transition, revealed the existence of the temperature interval where both phases are present simultaneously in the film (Figure 9c–f). The coexistence of two phases is a convincing indication of the first-order Sm-A–Hex-B transition in the 54COOBC films.

Moreover, similar spatially resolved maps of the 54COOBC films revealed that the temperature range of the two-phases coexistence, ΔT, is larger for thicker films and diminishes with the film thickness (Figure 10). This result is supported by earlier electron diffraction studies showing that in very thin (2 molecular layers) 54COOBC films, the Sm-A–Hex-B phase transition is continuous [71,72]. In Refs. [70,73] the observed effect was explained by an anomalously large penetration length $L_{0}$ of the surface hexatic order into the interior of the film in the vicinity of the tricritical point. This influences the temperature width of the two phase coexistence even in thick smectic films consisting of thousands of molecular layers [70,73]. Thus, by varying the thickness of the LC films, one can tune the proximity of the system to the tricritical point and eventually switch between the first- and second-order Sm-A–Hex-B phase transition.

### 4.2. Reconstruction of the Anisotropic Pair Distribution Function (PDF)

In any dense system, the positional and BO orders are coupled with each other, so for a complete description of the structure, one has to consider both of them. The most natural way for such a description is provided by the pair distribution function (PDF) [1,74], which in homogeneous systems can be determined as Equation (19):(19)$$g\left(r\right)=\frac{1}{{\langle n\rangle }^{2}V}\langle \underset{i\neq j}{\sum }\delta (r-r_{i}+r_{j})\rangle ,$$
where 〈*n*〉 is an average density of particles, V is a volume of the system, angular brackets 〈…〉 denote ensemble averaging, the indexes i and j enumerate all particles in the system, and $r_{i}$ is the position of the *i*-th particle. The PDF g(r) defines the probability of finding a particle at the separation r from any other arbitrarily chosen particle. Usually for isotropic systems, e.g., simple liquids, due to ensemble averaging in Equation (19) the angular dependence of g(r) washes out, so one considers the so-called radial distribution function g(r), which is the PDF g(r) averaged over all possible directions of the vector r. However, in the case of a system exhibiting the BO order, such angular averaging will lead to the loss of structural information. Thus, below we will consider non-averaged angular-dependent PDF.

If the PDF g(r) is invariant with respect to some symmetry transformations, the corresponding structure factor S(q) will also exhibit the same symmetry, as it directly follows from the following relation [1,75] Equation (20):(20)$$S\left(q\right)=1+\langle n\rangle \int \left(g\left(r\right)-1\right)e^{-iqr}dr.$$
In reference [76], the particular case of a 2D system was analyzed and the following relation between the angular Fourier components of the PDF $g_{n}\left(r\right)$ and the structure factor $S_{n}\left(q\right)$ was obtained Equation (21):(21)$$g_{n}\left(r\right)=\delta_{0,n}+\frac{1}{2\pi \langle n\rangle i^{n}}\overset{\infty }{\underset{0}{\int }}\left(S_{n}\left(q\right)-\delta_{0,n}\right)J_{n}\left(qr\right)qdq,$$
where $\delta_{0,n}$ is the Kronecker delta and $J_{n}\left(qr\right)$ is the Bessel function of the first kind of integer order n. The 2D PDF g(r)=g(r,θ), where r and θ are the polar coordinates of the 2D radius vector, can be evaluated as the angular Fourier series in Equation (22):(22)$$g\left(r,\theta \right)=\overset{\infty }{\underset{-\infty }{\sum }}g_{n}\left(r\right)e^{in\theta },$$
where the coefficients $g_{n}\left(r\right)$ are defined in Equation (21). Clearly, AXCCA, which is capable of accurate evaluation of the angular Fourier components of the scattering intensity, is well suited for the determination of $S_{n}\left(q\right)$ and subsequent reconstruction of the angular-resolved 2D PDF.

By applying this approach to the experimentally measured diffraction patterns of the 3(10)OBC LC compound, it was demonstrated how the sixfold rotational symmetry appears in the arrangement of molecules while the LC film goes throughout the Sm-A–Hex-B phase transition [76]. Figure 11 displays the direct correspondence between the sharpness of the hexatic diffraction peaks (Figure 11a–d) in reciprocal space and development of the sixfold BO order in molecular layers (Figure 11e–h).

The oscillations of the PDF in the Sm-A phase (Figure 11e) decay over few intermolecular distances, while in the Hex-B phase the magnitude of the peaks of the PDF stays almost constant over the same distance (Figure 11h). This indicates the simultaneous increase of the positional correlation length ξ with the development of the BO order. Another important observation is that the concentric rings of the PDF in the Sm-A phase (Figure 11e) gradually change into hexagonally arranged peaks in the Hex-B phase (Figure 11f–h), which is also an evidence of the coupling between the positional and BO orders. As a result of this coupling, the PDF decays at different rates in different directions (compare faster decay in the direction A and slower decay in the direction B in Figure 11h). In reference [76] it was shown that for an unambiguous determination of the positional correlation length in such systems with angular anisotropy one has to consider the decay of the projection of the PDF onto the direction of the diffraction peaks (direction A). In this case, the results obtained by measuring the width of the diffraction peaks in the radial direction, and analysis of the PDF in real space will give similar values of the positional correlation length ξ.

Another approach relevant to the 3D case was considered in reference [31], where AXCCA was used to reconstruct the orientational correlation function $\Theta \left(r,r^{\prime },\theta \right)$. This function determines the probability to find in the system two bonds of length r and $r^{\prime }$ with an angle θ between them. It has been theoretically shown that by measuring a large number (>$10^{5}$) of anisotropic diffraction patterns from liquid phases, metallic glasses, or organic molecules, one can in principle reconstruct the function $\Theta \left(r,r^{\prime },\theta \right)$.

### 4.3. Colloidal Systems

Another important and intensively studied class of soft matter system are colloidal systems, the unique properties of which, such as tunable photonic bandgap and response to external stimuli, make them attractive for many applications [77,78,79,80,81,82]. X-rays are especially well suited for structural studies of colloidal systems due to their high penetration length. However, the interpretation of the scattering data from bulk colloidal crystals possessing various defects is usually challenging. Using coherent X-ray diffraction, it was demonstrated that AXCCA is capable to detect rotational symmetry in a local arrangement of colloidal particles even in thick colloidal films [12]. In contrast to the previously discussed LC systems with the long-range BO order, in which the anisotropy of the diffraction patterns can be often easily identified by the unaided eye, the coherent diffraction patterns from colloids usually consist of the apparently random speckles along the scattering ring. Thus, AXCCA is in many cases a unique technique which allows one to study the local order in colloids [83].

In reference [84], X-ray diffraction with a 400 nm beam was combined with AXCCA to study the structure of a dry colloidal film consisting of spherical silica nanoparticles of 19.3 and 12.2 nm in diameter. Applying AXCCA to individual diffraction patterns and collecting data at different spatial positions on the sample, the authors were able to observe changes of the colloidal crystal structure over the scanned area. In Figure 12, the spatially resolved maps of two angular Fourier components, evaluated by AXCCA as $\left|{\hat{I}}_{l}\left(Q\right)\right|=\sqrt{{\hat{C}}_{l}\left(Q\right)}$, where ${\hat{C}}_{l}\left(Q\right)$ are the corresponding Fourier components of the normalized CCF, are shown. The signal at $Q_{1}=0.16\;{\mathrm{nm}}^{-1}$ and $Q_{2}=0.31\;{\mathrm{nm}}^{-1}$ corresponds to the scattering from larger and smaller particles, respectively. The maps of higher order components l=4, 6 show how the BO order changes depending on the local ratio between the number of large and small particles (Figure 12c–f).

Similar spatially resolved maps were retrieved for the phase $\Omega_{l=4}$ of the normalized intensity angular Fourier components ${\hat{I}}_{l}\left(Q\right)=\left|{\hat{I}}_{l}\left(Q\right)\right|exp\left(i\Omega_{l}\left(Q\right)\right),$ which can be evaluated as the angular Fourier coefficients of the scattering intensity along the *Q*-ring (see Equation (8)). This phase quantifies the orientation of the colloidal crystal structure with respect to a reference axis. Two maps at different *Q*-values in Figure 12 g,h show the existence of patches of similar orientations within the sample, which resemble the drying front of silica particle suspensions. An analogous approach which includes a combination of the focused X-ray beam and AXCCA was used in [19] to investigate patches with four- and sixfold orientational order in the film of self-assembled gold nanoparticles.

In some cases of nearly crystalline structures, it is viable to consider the initial angular CCF rather than its Fourier components. The colloidal crystals made of poly(methyl-methacrylate) (PMMA) hard-spheres of 125.5 nm in diameter were studied in [85]. In Figure 13, well distinguishable peaks of the angular CCF measured at four different *q*-values indicate that the sample is crystalline. To decide between two preferable close-packed structures, face-centered cubic (fcc) or hexagonal close-packed (hcp), the model CCFs for these two structures were evaluated. For each structure (fcc or hcp), 10^3^ perfect crystals in random orientations were modeled for monodisperse particles of the same size and volume fraction as the sample system in the experiment, and the CCFs at the same *q*-values were calculated for each diffraction pattern and averaged. A comparison between the experimental and model data (Figure 13) has revealed that the structure of the real colloidal crystals is more likely to be fcc with indications of stacking faults. We should note here that in this case the standard analysis of the measured angular averaged structure factor of the colloidal crystal S(q) fails to distinguish between fcc and hcp structures. Only additional information about the angular correlations accessible by AXCCA can overcome the limitation of powder average and enable access to the details of the crystal structure.

While the standard analysis of angular averaged scattering intensity shows excellent results for isotropic systems, like liquids and powders, and various crystallographic techniques are well suited to study the structure of crystals, AXCCA appears to be very useful to study the systems with weak orientational order. One of the prominent examples of such systems is a suspension of colloidal particles. The aqueous solution of charge-stabilized polyacrylate colloidal particles was studied at high pressure in reference [86]. X-ray diffraction patterns at ambient and high pressures (p=1 bar and p=2500 bar, respectively) in Figure 14a,b clearly show the development of the sixfold BO order in the system. The evolution of the angular averaged structure factor S(Q) as a function of increasing pressure also indicates a transition from isotropic liquid-like structure of the colloidal suspension to the fcc structure Figure 14c. These observations were supported by the increasing magnitude of the sixfold modulations of the CCF (Figure 14d). Surprisingly, AXCCA also revealed fourfold orientational order in the system under ambient pressure, which is not well resolved in Figure 14a. The authors attributed this fourfold symmetry to the metastable ordered state of the colloidal suspension at ambient pressure due to shear-induced melting.

## 5. Mesocrystals Formed by Nanoparticles

Superstructures of nanoparticles, which preserve the unique properties of nanoparticles, and at the same time allow easy integration into devices due to their mesoscopic structure, have attracted increasing interest in the last decade [87,88,89,90,91]. In this section, we will focus on the studies of a specific type of such superstructures, called mesocrystals, consisting of periodically arranged iso-oriented nanoparticles spatially separated by some media [92]. Inorganic nanoparticles interlinked with organic π-systems are particularly promising materials for various applications, including field-effect transistors (FETs), light-emitting diodes (LEDs), photodiodes, and photovoltaic cells (PVCs) [93,94]. Such organic-inorganic mesocrystals can be conveniently fabricated by self-assembly of nanoparticles from solution by exploiting ligand–ligand interactions [95,96,97]. However, a self-assembled superlattice is prone to various defects, which influence the electrical, optical, and mechanical properties of the mesocrystals [98,99,100]. Nanofocused X-ray diffraction in combination with AXCCA is a unique tool allowing one to explore the real structure of mesocrystals, which should facilitate the applications of synthetic mesocrystals.

In contrast to colloidal crystals consisting of spherical particles, which were discussed in the previous section, the nanoparticles here are faceted single crystals, and therefore exhibit significant anisotropy [101,102]. Facet-specific interaction between organic ligands and inorganic nanoparticles can lead to a well-defined orientation of the nanoparticles with respect to the mesocrystalline superlattice [103,104,105]. Thus, a comprehensive structural study of mesocrystals addresses not only the structure of the superlattice, but also the orientation of the nanoparticles with respect to the superlattice. AXCCA appears to be especially suited for such studies since it naturally allows to investigate mutual angular correlations between the mesocrystal superlattice and the atomic lattice of the nanoparticles by means of the CCF $\hat{C}\left(q_{1},q_{2},\Delta \right)$.

The structure of a mesocrystalline assembly of PbS nanoparticles with a diameter of 6.2 nm in size, interlinked with the organic π-system tetrathiafulvalenedicarboxylate (TTFDA), was studied in reference [18] (Figure 15a,b). Approximately 200–300 nm thick mesocrystals were scanned with the X-ray beam focused down to 400×400 nm^2^. A large 2D detector positioned behind the sample was shifted from the optical axis of the beam to record simultaneously the small-angle X-ray scattering (SAXS) from the superlattice at q~1–2 nm^−1^ (Figure 15c) and the wide-angle X-ray scattering (WAXS) signal from the atomic lattice (AL) of individual PbS nanoparticles. As a result, it was possible to measure the scattering signal at the scattering vector $q_{111}^{AL}=18.3$ nm^−1^ and $q_{200}^{AL}=21.2$ nm^−1^, corresponding to the 111 and 200 reflections from the PbS atomic lattice of the nanoparticles (Figure 15c). The presence of well-defined reflections from the PbS atomic lattice clearly indicates that, at least within the illuminated volume, the nanoparticles have a preferred orientation with a mean deviation (“mosaicity”) ΔΦ about 10°.

To investigate the structure of the mesocrystal on a larger scale, maps of the orientation of the diffraction peaks were analyzed. In Figure 16a, green arrows represent the angular position of the strongest superlattice (SL) peaks at $q_{112}^{SL}=1.72$ nm^−1^, which were attributed to the {112} reflections. A similar map of the angular positions of the {111} and {200} reflections from the PbS atomic lattice of nanoparticles is shown in Figure 16b. A comparison between these two maps reveals the presence of domains with typical linear dimensions of 2–4 μm, in which the orientation of both, the superlattice and the individual nanoparticles, is preserved.

For the quantitative characterization of the mutual orientation of the nanoparticles with respect to the superlattice, the CCF $C\left(q_{111}^{AL},q_{112}^{SL},\Delta \right)$ was evaluated to verify if the angular positions of the diffraction peaks from the atomic lattice in WAXS are correlated with the angular positions of the diffraction peaks from the superlattice in SAXS. The CCF averaged over all diffraction patterns, where both WAXS and SAXS signals are present, is shown in Figure 16c. Well-distinguishable peaks of the CCF clearly indicate that a certain orientation of the nanoparticles in the superlattice is preserved, not only within the domains but in all scanned regions of the sample. Further analysis of the CCF, which required simulations of the scattering patterns and evaluation of the model CCF shown in Figure 16d, revealed that the atomic lattice and the superlattice of the mesocrystal have five common directions, namely three 〈100〉 directions, as well as the [110] and [−110] directions (Figure 16e). Moreover, the sensitivity of AXCCA to angular positions of the scattering peaks allows one to establish that the mesocrystal superlattice exhibits a tetragonal distortion with c/a≈1.22 (a=7.9 nm, c=9.7 nm), which appears as splitting of the CCF peaks at about Δ = ±90° (Figure 16c,d). The width of the CCF peaks also contains important information about the orientational order in the system, namely that the above-mentioned mutual orientation of the nanoparticles and superlattice is preserved in all scanned positions with a mean deviation of ~2.5°.

The details of attachment of organic ligands to the nanoparticles were addressed in reference [106], where electron diffraction was used. The authors studied a monolayer of 1-dodecanethiol-coated gold nanoparticles of 5.7 nm in diameter. Electron diffraction allows collecting sufficient scattering signal from light atoms (H, C, S) in the organic ligands and hence be sensitive to the orientation of the rod-like organic molecules with respect to the facets of the nanoparticles. The angular CCF C(s,Δ) was evaluated along the three scattering rings at $s=s_{1},\;s_{3},\;s_{4}$, corresponding to different length scales in real space. Thus, the scattering signal at $s_{1}=0.66$ Å^−1^ corresponds to the inter-particle distance $d_{1}=6.6$ nm in the close-packed hexagonal structure (Figure 17a). The Fourier spectra of the CCF at $s=s_{1}$ contain dominant contributions from the second, fourth and sixth order components (Figure 17b). The ν=2 and ν=4 components were attributed to the astigmatism of the electron beam and scattering from the substrate. After subtraction of these contributions from the experimental CCF, the resulted function exhibits a clear sixfold symmetry which was assigned to the hexagonal superlattice (Figure 17c).

The scattering signal at $s_{3}=1.45$ Å^−1^ corresponds to the distance $d_{3}=4.43$ Å between two nearest locations of the attached ligands (Figure 17d). After filtering the ν=2 and ν=4 components (Figure 17e), the experimental CCF shows no sixfold or higher rotational symmetry (Figure 17f), which is explained by the fact that ligands are attached to all facets of the nanoparticles, so the scattering signal at $s_{3}$ is averaged over all angles. The signal at $s_{4}=1.68$ Å^−1^ corresponds to the real-space distance $d_{4}=3.72$ Å between parallel ligands, which occurs when the ligands are attached to the facets at a certain angle (Figure 17g). The Fourier analysis of the corresponding CCF at $s_{4}$ reveals a strong contribution of the ν=4 and ν=6 components (Figure 17h,i). This effect was explained by two factors: (1) The nanoparticles assemble in a hexagonal or tetragonal superlattice and (2) a preferred orientation of the nanoparticles with respect to the superlattice. Thus, even without recording the diffracted signal from the atomic lattice of gold nanoparticles, the analysis of the angular correlations allowed the authors to establish a certain degree of orientational order between individual nanoparticles as well as obtaining information on how the organic ligands are attached to the facets of the nanoparticles.

### Laser-Induced Orientational Order in Photo-Reactions

In the previous sections we discussed the studies of the orientational order in the systems in equilibrium. The recently developed XFELs are capable to produce femtosecond X-ray pulses that allow to probe fast dynamics at nanoscale, for example, molecular dynamics. The potential of AXCCA to study the evolution of the orientational order in the time-resolved X-ray scattering experiment on solvated molecules was demonstrated in reference [58]. In this work, the aqueous molecules of tetrakis-μ-pyrophosphitodiplatinate (II) (PtPOP) were excited by a short 50 fs optical laser pulse (pump) with a wavelength of 395 nm. From the whole ensemble of randomly oriented PtPOP molecules, the linearly polarized optical light selectively excites only the molecules, in which the Pt-Pt axis is oriented parallel to the external laser electric field ***E*** (Figure 18a). The succeeding 50 fs 9.5 keV X-ray pulse (probe) produces a snapshot of the configuration after photo-excitation at a single time-delay τ (see scheme of the setup in Figure 18b).

The change of the electronic structure in the excited PtPOP molecules leads to contraction of the Pt-Pt bond approximately by 0.24 Å. This contraction can be readily detected on the difference signal images which are created by subtracting an X-ray diffraction pattern from non-excited molecules in the ground state from an X-ray diffraction pattern from the partially excited ensemble of molecules. Thus, the difference diffraction images contain only the signal from changes in the PtPOP molecules upon excitation and their interaction with the solvent. To study the dynamics of the photoexcited molecules, approximately N=3000 difference diffraction patterns were collected at each time bin of 1 ps. Applying AXCCA, it was shown that the difference diffraction signal is anisotropic, and moreover, the anisotropy can be well described by the second Fourier component $C_{2}\left(q\right)$ (see Equation (12)), which is in agreement with the theoretical approach proposed in reference [107].

The temporal evolution of the difference diffraction pattern anisotropy, i.e., temporal decay of the value of $\langle C_{2}\left(q\right)\rangle$, is shown in Figure 18c. Fitting the total decay by a sum of two exponential terms, the authors found two different time scales occurring in the system. The longer time constant of 46±10 ps corresponds to the rotational dephasing of the initial angular distribution of the molecules, and the shorter time constant of 1.9±1.5 ps may be attributed to the dampening time for Pt–Pt bond vibrations.

## 6. Conclusions and Outlook

Soft matter materials are ubiquitous and have found many applications in modern technology. Usually the structural studies of soft matter systems include an analysis of the pair correlation functions between particles in the system. However, the development of experimental techniques and theoretical approaches revealed the importance of the mutual arrangement and orientation of the particles on the properties of the system. The importance of the orientational order is well-known for LCs, in which a specific orientation of anisometric molecules leads to the formation of numerous LC phases with different optical, electrical and mechanical properties.

Another type of orientational order, so-called bond-orientational order, is related not to the orientation of individual particles, but to the orientation of imaginary “bonds” between them. First discovered in a process of 2D crystal melting, it was realized later that the BO order can also exist in 3D systems. These includes the so-called stacked hexatic phase, where the BO order exists across the 2D molecular layers, and also “true” 3D systems, for example colloids, glasses and quasicrystals, in which particles form local 3D clusters of certain symmetry. Nowadays, such complex materials as mesocrystals can be synthesized, which can be seen as a superlattice formed by oriented nanoparticles. The orientation of the atomic lattice of individual nanoparticles with respect to the crystallographic directions of the superlattice plays an essential role in the electrical properties of mesocrystals. The appearance of such new complex materials with intricate relations between the structure and functional properties require the development of new methods for characterization of these systems.

AXCCA is a basic analysis tool which is capable to quantitatively characterize the angular anisotropy of the X-ray diffraction patterns and relate this anisotropy to the structural properties of the system under investigation. Originally, AXCCA was used for the analysis of SAXS diffraction patterns from solution of bioparticles to obtain additional information on the structure of an individual particle [8,50,57]. Later it was demonstrated that a combination of AXCCA with a coherent, focused X-ray beam can reveal local orientational order in colloidal crystals [12], which gave rise to intensive studies of various dense systems by AXCCA [15,17,18,19]. The combination of AXCCA with X-ray microscopy using a micro- or even nanofocused X-ray beam gives information not only on the local order in the system, but also reveals how this local order changes depending on the spatial position in the system.

An intriguing option is to combine AXCCA with ultra-short X-ray pulses from XFEL sources. This allows one not only to study the static structure of the system, but also to investigate dynamical processes. In order to measure the coherent diffraction from a dynamical system, the X-ray pulse duration should be shorter than the characteristic time scale in the system; otherwise the speckles on the diffraction pattern would smear out due to the movement of the particles. This also means that the incoming photon flux should be high enough to produce sufficient diffraction signal from a single shot. The usage of short X-ray pulses together with AXCCA potentially opens the opportunity to study the structure of molecular clusters which are instantaneously formed in liquids, for example, in water [108]. The recent paper by P. Vester et al. [58] sets a benchmark for AXCCA to provide an asset in such time-resolved studies of molecular dynamics at XFELs. We foresee that it is only a matter of time before AXCCA is combined with the techniques studying dynamics of the systems such as X-ray photon correlation spectroscopy (XPCS).

Despite the undoubted progress in extracting structural information using AXCCA, there are a few aspects which can be addressed in future. First of all, similar to many other X-ray techniques, one of the limiting factors are X-ray detectors. For AXCCA, it is desirable to record the full scattering ring in order to perform the Fourier analysis of the angular anisotropy of the diffracted intensity. The smaller the length scales of the scattering objects, the larger are the X-ray scattering angles, which means that larger detectors are required to capture the whole scattering ring. Sometimes it is sufficient to record only a part of the scattering ring (see, for example, [18,99]), but this usually requires some pre-existing knowledge about the structure of the system, so this approach cannot be used in general. Another possible improvement could be the development of faster detectors. The existing 2D detectors allow collecting a limited number of images with MHz frequency, however in a continuous regime the frame rate is limited by kHz. Increase of the frame rate would allow combining of AXCCA with XPCS type measurements to investigate the dynamics of various soft matter systems in the non-destructive regime.

In the interpretation of the AXCCA results, the most complicated part is to relate the anisotropy of the measured diffraction patterns to the actual structure of the system, especially in the 3D case. One of the possible ways to simplify this problem is to limit the number of illuminated particles, which can be obtained by focusing of the incoming beam. Focusing would also allow to perform a finer raster scan of the sample, which can provide an important structural information on the inhomogeneity of the sample, as it was demonstrated in many examples of this review. However, due to the low efficiency of the X-ray focusing optics, the high initial flux is required in order to have sufficient number of X-ray photons in the focused beam. Moreover, these photons have to be coherent in order to create the speckles on the diffraction pattern.

Fortunately, the brightness of the modern X-ray sources already provides a sufficient flux of coherent photons for studies of the soft matter. Nowadays, the most advanced diffraction limited storage rings (DLSR) are specially designed to supply the highest possible coherent photon flux. Currently, the first multiband-achromat synchrotron source MAX IV (Lund, Sweden) is already under operation, and several more synchrotron sources are under commissioning, upgrade or in the planning stage. These are, for example, SIRIUS (Campinas, Brazil), ESRF-EBS (Grenoble, France), APS-U (Argonne, USA), and PETRA IV (Hamburg, Germany). The higher degree of coherence from these new sources would increase the contrast of the diffraction patterns, which would result in a higher signal-to-noise ratio.

Finally, we would like to speculate on the future development of AXCCA. Being a general analysis technique, AXCCA could be used in studies of almost any system with angular correlations. In this review, we considered several examples, such as the hexatic phase in LCs, colloidal crystals and self-assembling nanoparticles, but in principle AXCCA could provide valuable information on the orientational order in almost any soft matter system. Moreover, combining AXCCA with resonant elastic X-ray scattering (REXS), could potentially reveal hidden features of magnetic and charge order superstructures in hard condensed matter systems, for example, the lattice of skyrmions [109] or ferroelectric domain walls [110]. It was also shown [111] that combinations of AXCCA with the multiple-wavelength resonant scattering can be used for element-specific imaging of various nanoscale objects.

We also would like to point out that the general mathematical apparatus of AXCCA can be extended beyond X-ray scattering. Usage of electrons as a probe for scattering allows to address problems which can hardly be solved by X-ray diffraction. This includes very thin or weakly scattering samples almost transparent for X-ray, for example very thin LC films [72], monolayer superlattices of nanoparticles [106], or metallic glasses [112,113].

AXCCA has been developed to investigate the structure of partially ordered systems, to detect hidden symmetries and weak angular correlations. These include various complex fluids, polymer melts, colloids, liquid crystals, suspensions of biological molecules, block copolymers and mesocrystals. Having in mind the astonishing achievements in the synthesis of new molecules of almost any shape and sequences, there are no doubts that in the near future a great number of new substances will appear, the properties of which are hard to imagine at the present time. The various types of orientational order should be an essential feature of newly synthesized materials. Thus, the role of AXCCA as a unique tool in X-ray studies of orientational order and angular correlations will only increase with time.

## Figures and Tables

**Figure 1 materials-12-03464-f001:**
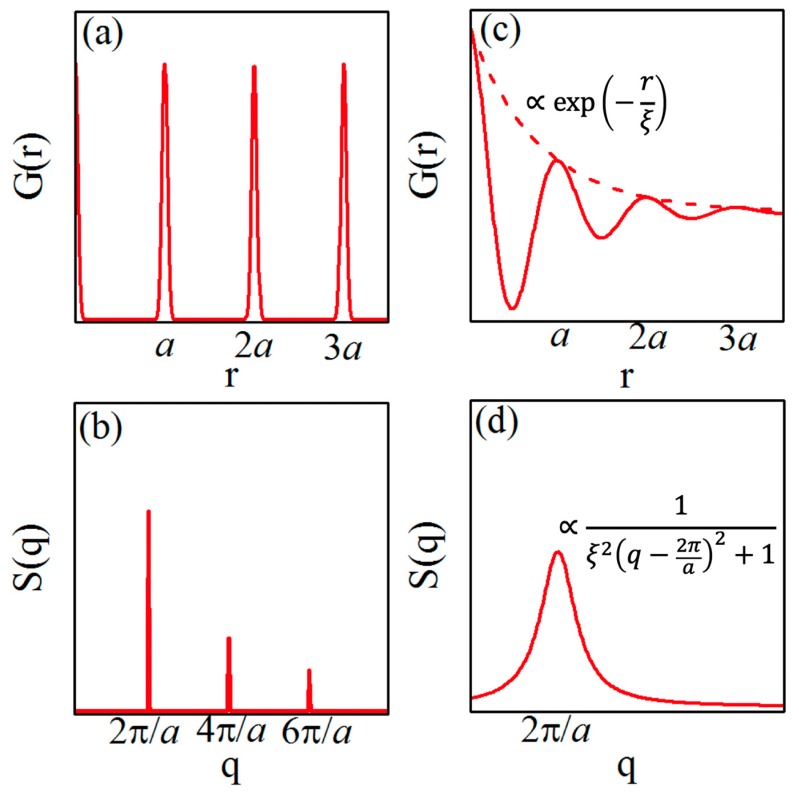
Schematic representation of the correlation function G(r) in real space and the structure factor S(q) in reciprocal space for one-dimensional systems with long-range order (**a**,**b**) and short-range order (**c**,**d**). Due to the exponential decay of the envelope of the correlation function in the case of short-range order (**c**), only one broad Lorentzian-shape peak is visible in the structure factor (**d**). (Adapted from de Jeu et al. [20]). Reproduced with permission of American Physical Society.

**Figure 2 materials-12-03464-f002:**
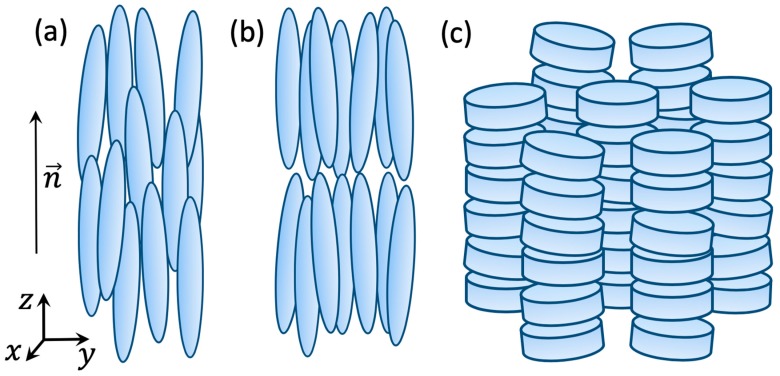
Schematic representation of different LC phases. (**a**) The nematic phase, in which elongated molecules are preferably aligned along the director **n**; (**b**) The smectic-A phase, in which elongated molecules form parallel layers in addition to the nematic orientational order; (**c**) The columnar phase, in which aligned disc-like molecules form 2D lattice of columns.

**Figure 3 materials-12-03464-f003:**
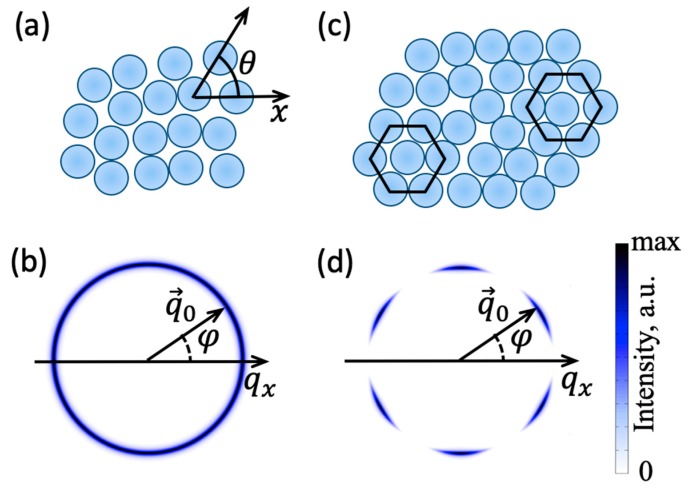
(**a**) Local BO order field $\Psi_{n}\left(r\right)$ in a 2D system with the short-range BO order; (**b**) Diffraction pattern in the case of the short-range bond-orientational (BO) order; (**c**) schematic structure of the 2D hexatic phase exhibiting a quasi-long-range sixfold BO order; (**d**) diffraction pattern from the 2D hexatic phase. Color in (**b**,**d**) represents the intensity of the scattered X-rays.

**Figure 4 materials-12-03464-f004:**
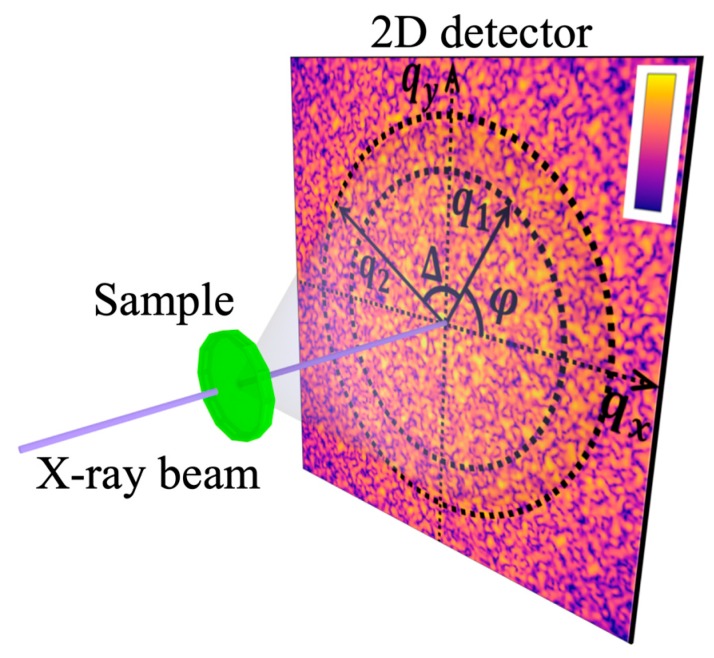
Scheme of an X-ray diffraction experiment in transmission geometry. The focused X-ray beam can probe different spatial positions on the sample. Diffraction pattern recorded by a 2D detector positioned behind the sample and polar coordinate system for evaluation of the angular cross-correlation function $C\left(q_{1},q_{2},\Delta \right)$ are shown. The intensity of the diffraction pattern is coded by color.

**Figure 5 materials-12-03464-f005:**
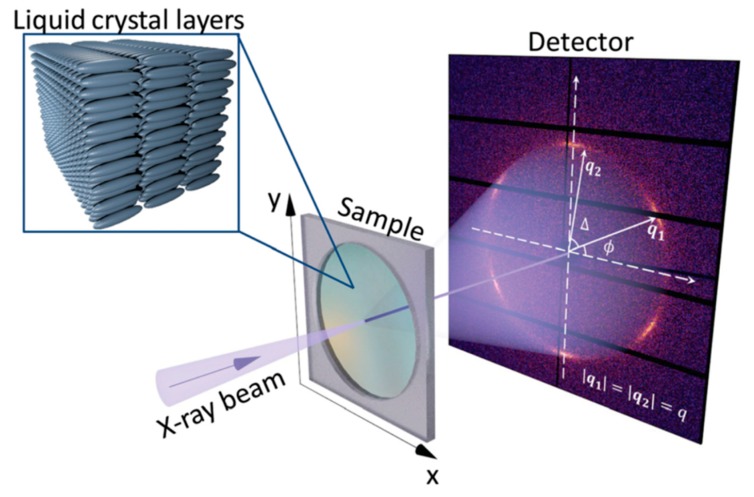
Scheme of the diffraction experiment on free standing liquid crystal (LC) films. The molecular layers are oriented parallel to the film surface, so the X-ray diffraction pattern corresponds to the in-plane order. (adopted from Zaluzhnyy et al. [17]—Published by The Royal Society of Chemistry). Used under CC-BY 3.0.

**Figure 6 materials-12-03464-f006:**
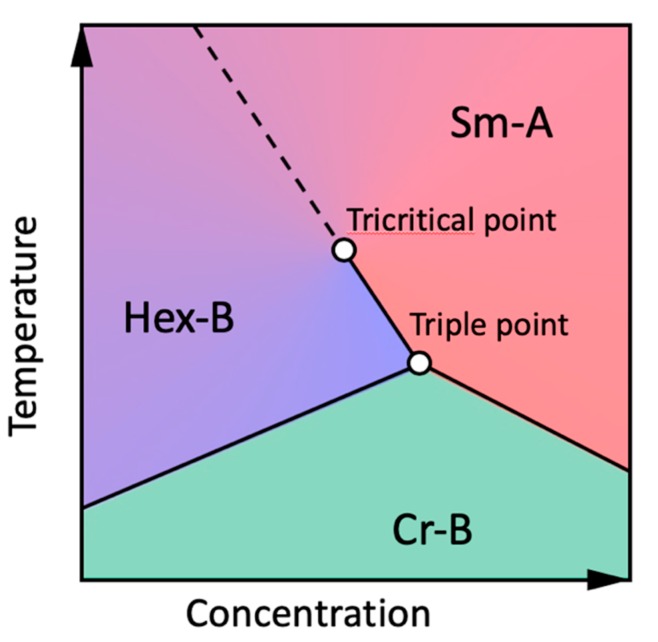
Generic phase diagram of the LC in the vicinity of the Hex-B–Sm-A–Cr-B triple point for a 3D case according to [40]. Solid and dashed lines correspond to the first- and second-order phase transitions, respectively (adopted from Zaluzhnyy et al. [17]––Published by The Royal Society of Chemistry). Used under CC-BY 3.0.

**Figure 7 materials-12-03464-f007:**
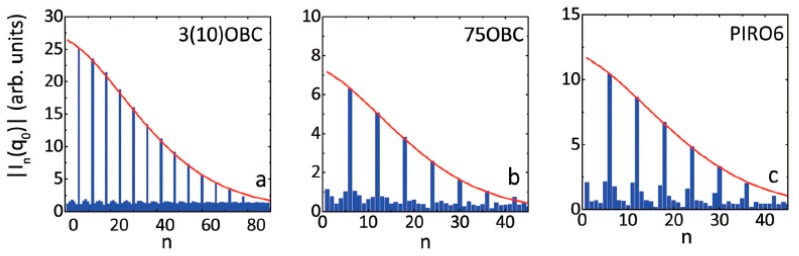
Magnitudes of the Fourier components $\left|I_{n}\left(q_{0}\right)\right|$ as a function of the order number *n* for three different compounds: 3(10)OBC at the relative temperature $\Delta T=T-T_{C}=-5.3$ °C (**a**), 75OBC at ΔT=−4.55 °C (**b**) and PIRO6 at ΔT=−4.3 °C (**c**). The temperature ΔT is calculated relative to the respective temperature $T_{C}$ of the second-order Sm-A–Hex-B phase transition for each compound. Red lines show fitting using the MCST (adopted from Zaluzhnyy et al. [17]––Published by The Royal Society of Chemistry). Used under CC-BY 3.0.

**Figure 8 materials-12-03464-f008:**
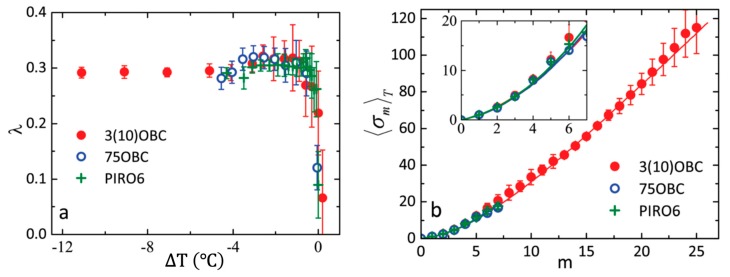
(**a**) Temperature dependence of the parameter λ for three LC compounds, obtained from the fits of experimental data with scaling relations of multicritical scaling theory (MCST). The relative temperature $\Delta T=T-T_{c}$ is measured from the Sm-A–Hex-B phase transition temperature $T_{C}$; (**b**) Temperature averaged values of ${{\langle \sigma }_{m}\rangle }_{T}$ and their fit with the MCST for three compounds. In the inset, an enlarged region for m values from one to seven is shown. (adopted from Zaluzhnyy et al. [17]–Published by The Royal Society of Chemistry) Used under CC-BY 3.0.

**Figure 9 materials-12-03464-f009:**
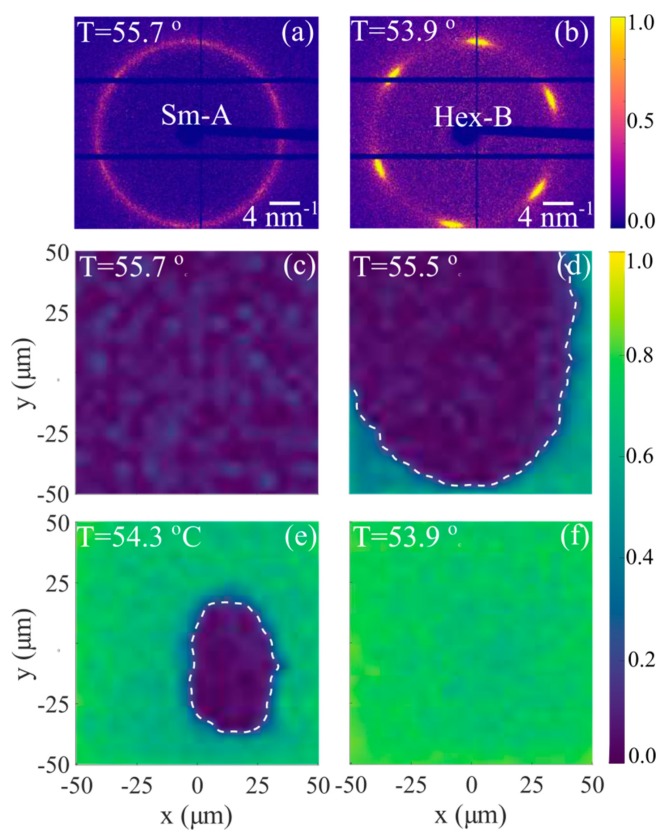
Coexistence of the smectic-A (Sm-A) and stacked hexatic (Hex-B) phases in a 54COOBC compound. (**a**,**b**) Examples of the in-plane diffraction from the Sm-A phase (**a**) and Hex-B phase (**b**); (**c**–**f**) spatially resolved maps of the BO order parameter C_6_. The color code indicates the normalized intensity of the scattered X-rays. The white dashed line in (**d**,**e**) separates regions of Sm-A (blue) and Hex-B (green) phases. (adopted from Zaluzhnyy et al. [70]). Reproduced with permission of American Physical Society.

**Figure 10 materials-12-03464-f010:**
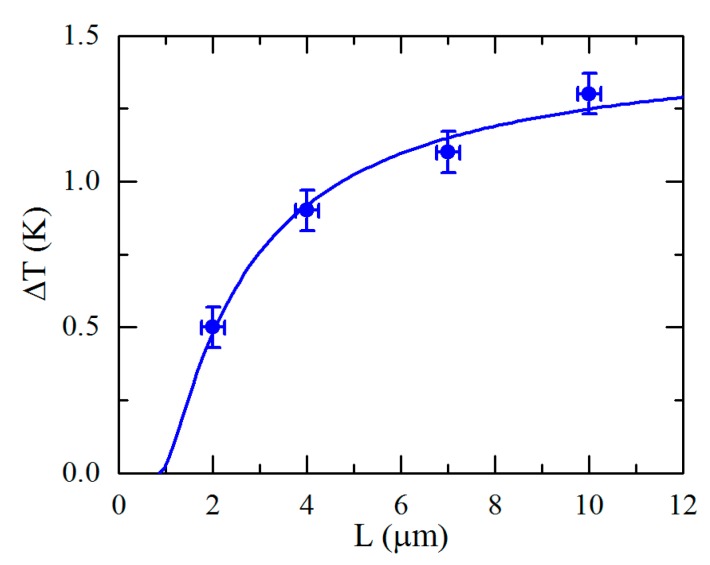
Dependence of the temperature width of the Sm-A and Hex-B coexistence region on the LC film thickness. The solid line shows the result of fitting with the analytical equation $\Delta T\propto {\left(1-L_{0}/L\right)}^{2}$, where $L_{0}\approx 0.9$ μm is the characteristic length scale (adopted from Zaluzhnyy et al. [70]). Reproduced with permission of American Physical Society.

**Figure 11 materials-12-03464-f011:**
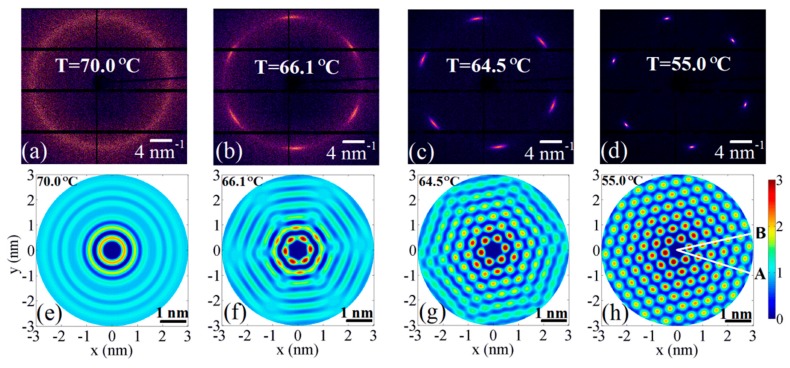
(**a**–**d**) Measured diffraction patterns from 3(10)OBC LC compound above (**a**) and below (**b**–**d**) the Sm-A–Hex-B phase transition. (**e**–**h**) Corresponding reconstruction of the 2D pair distribution function (PDF). Development of the BO order and increase of the positional correlation length is visible in (**f**–**h**). (adopted from Zaluzhnyy et al. [76]). Reproduced with permission of American Physical Society.

**Figure 12 materials-12-03464-f012:**
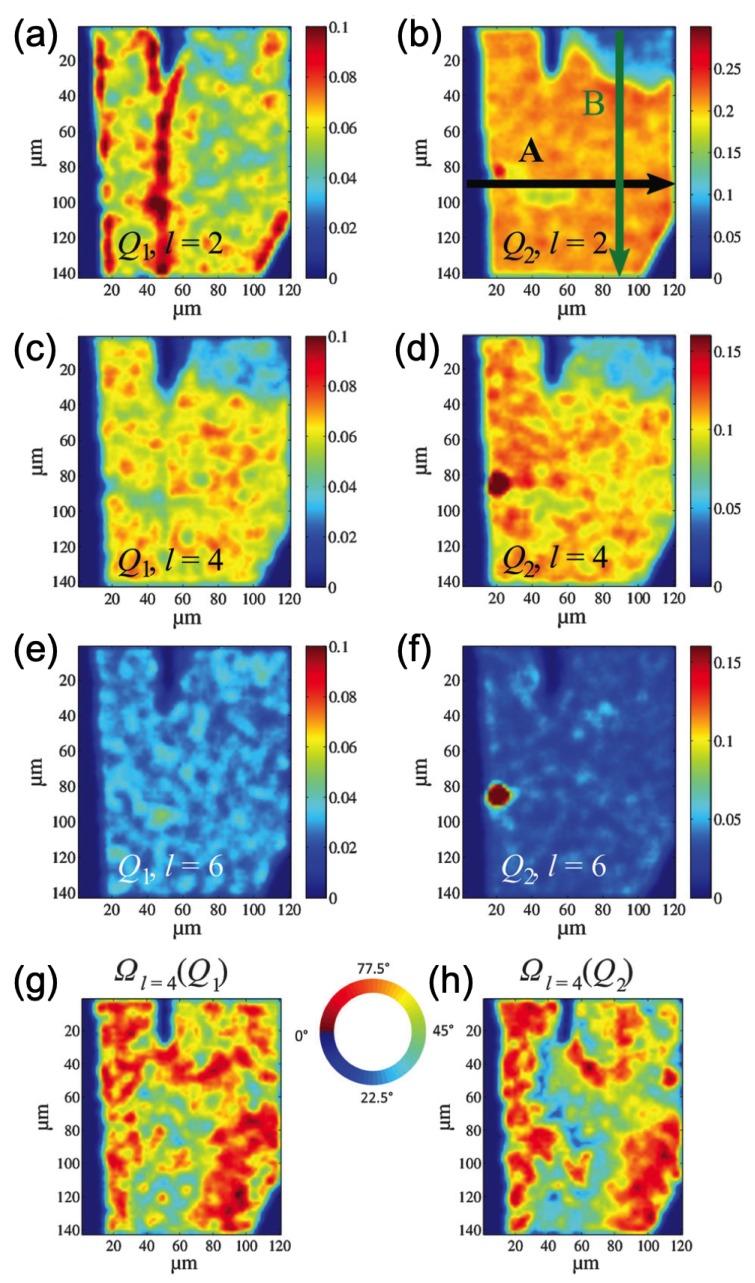
(**a**–**f**) Spatially resolved maps of the magnitude of the normalized intensity Fourier components $\left|{\hat{I}}_{l}\left(Q\right)\right|$ of the colloidal film for l = 2, 4, 6 at $Q=Q_{1}$ (**a**,**c**,**e**) and $Q=Q_{2}$ (**b**,**d**,**f**). (**g**,**h**) Spatially resolved maps of the phase $\Omega_{l=4}$ of the normalized intensity angular Fourier components ${\hat{I}}_{l=4}\left(Q\right)$ for $Q=Q_{1}$ (**g**) and $Q=Q_{2}$ (**h**) (adopted from Shroer et al. [84]—Published by The Royal Society of Chemistry). Used under CC-BY 3.0.

**Figure 13 materials-12-03464-f013:**
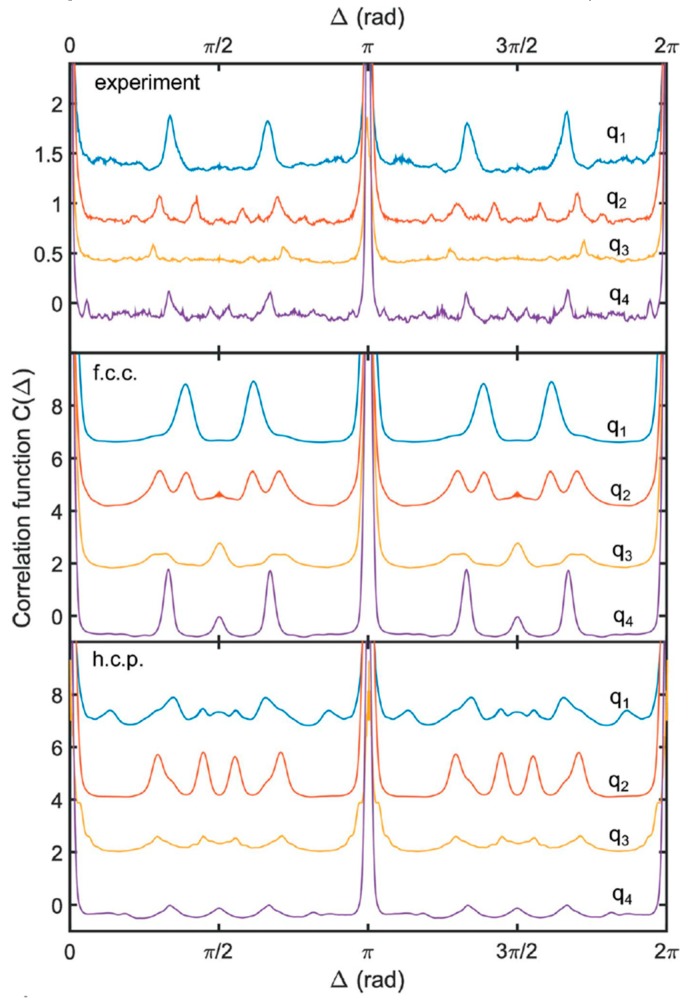
Experimentally measured cross-correlation function (CCF) C(q,Δ) at four different values of the momentum transfer $q_{1}=0.027$ nm^−1^, $q_{2}=0.029$ nm^−1^, $q_{3}=0.031$ nm^−1^, $q_{4}=0.044$ nm^−1^ (top) and modelled CCFs assuming ideal face-centered cubic (fcc) (middle) and hexagonal close-packed (hcp) (bottom) structures. Curves are shifted vertically for better visual appearance. (adopted from [85]). Used under CC-BY 4.0.

**Figure 14 materials-12-03464-f014:**
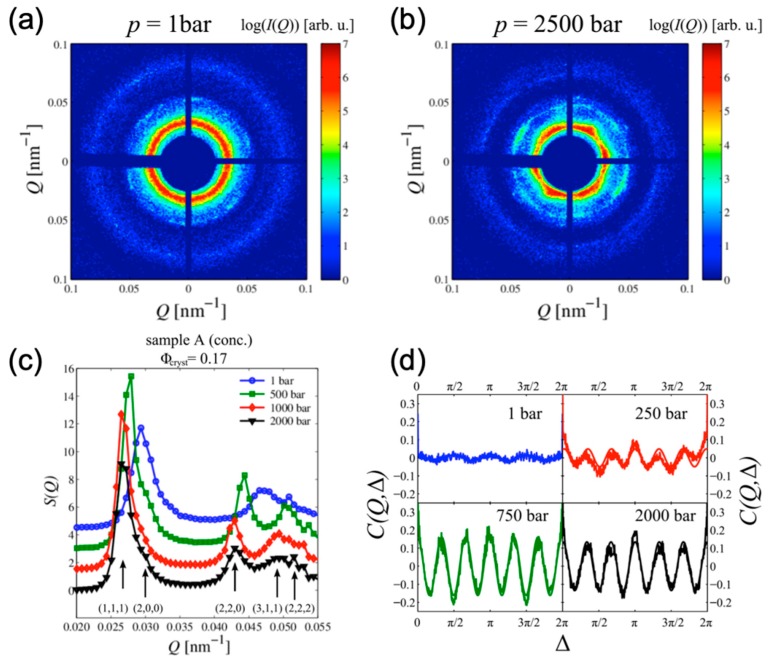
(**a**,**b**) Diffraction patterns from solution of charge-stabilized polyacrylate particles suspended in water at ambient pressure (**a**) and under pressure of 2500 bar (**b**). (**c**) Evolution of the structure factor under applied pressure. (**d**) Evolution of the cross-correlation function C(Q,Δ) under increasing pressure directly reveals the development of BO order (reprinted from Schroer et al. [86], with permission of AIP Publishing).

**Figure 15 materials-12-03464-f015:**
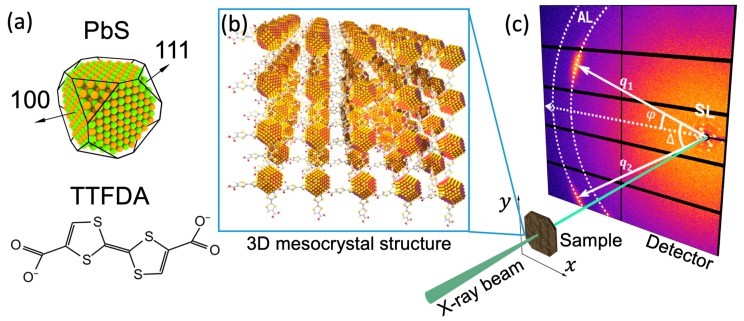
(**a**) Schematic view of a single PbS nanoparticle with the shape of a truncated cube with six 100 and eight 111 facets, and the chemical structure of the tetrathiafulvalenedicarboxylate (TTFDA) anion; (**b**) Structure of a mesocrystal formed by PbS nanoparticles crosslinked with TTFDA molecules; (**c**) Scheme of the diffraction experiment, in which small angle scattering from the mesocrystal superlattice and wide angle scattering from PbS atomic scattering were recorded simultaneously by a large 2D detector. (Adapted with permission from Zaluzhnyy et al. [18]. Copyright (2017) American Chemical Society).

**Figure 16 materials-12-03464-f016:**
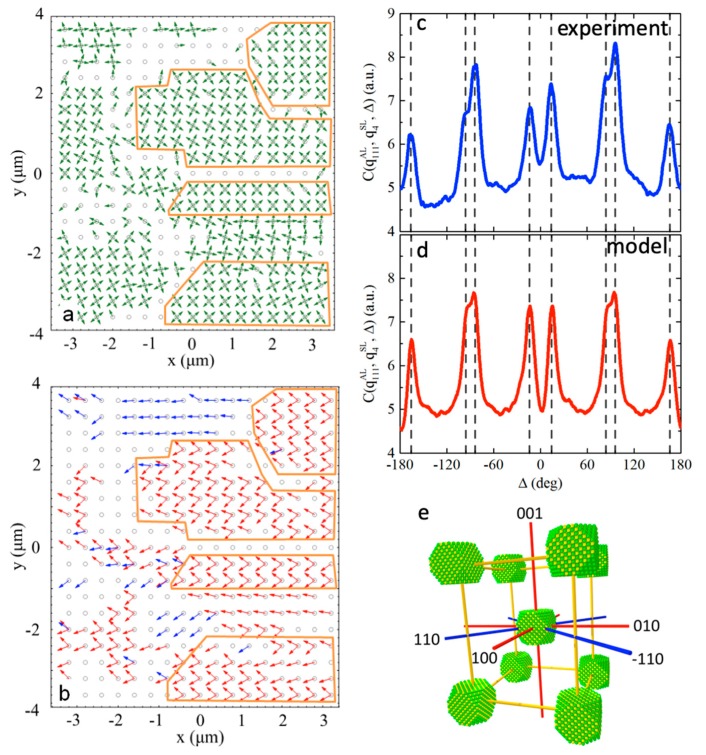
(**a**,**b**) Spatially resolved maps, showing the angular positions of the {112}_SL_ reflections of the superlattice (green arrows in (**a**)); {111}_AL_ and {200}_AL_ reflections of the atomic lattice (red and blue arrows in (**b**); respectively). Single-crystalline domains with a persistent orientation of PbS particles with respect to the superlattice are marked in orange; (**c**,**d**) Experimentally measured (**c**) and modelled (**d**) cross-correlation functions $C\left(q_{112}^{SL},q_{111}^{AL},\Delta \right)$ are shown by blue and red color, respectively; (**e**) Schematic of the mesocrystal unit cell displaying the angular correlation between the atomic lattice (AL) and superlattice (SL); collinear axes are indicated in red (〈100〉 directions) and blue (〈110〉 directions) (Adapted with permission from Zaluzhnyy et al. [18]. Copyright (2017) American Chemical Society).

**Figure 17 materials-12-03464-f017:**
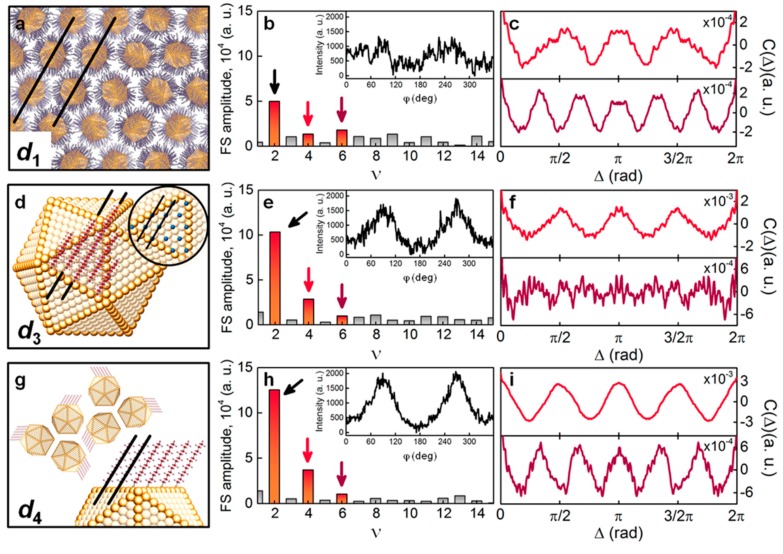
(**a**,**d**,**g**) Illustration of the three feature length scales in mesocrystals: $d_{1}\approx 6.6$ nm corresponds to the distance between Au nanoparticles (**a**), $d_{3}\approx 0.433$ nm corresponds to the hexagonal superlattice of dodecanethiol ligands bound to the gold core facets at specific locations between gold atoms (**d**), $d_{4}\approx 0.372$ nm corresponds to the distance between carbon atoms of ordered ligand chains (**g**). (**b**,**e**,**h**) Magnitudes of the angular Fourier components of intensity $I_{\nu }\left(s\right)$ at scattering vectors *s*_1_, *s*_3_ and *s*_4_ corresponding to the distances *d*_1_, *d*_3_ and *d*_4_, respectively. The angular dependence of the scattered intensity is shown in the insets by black lines. (**c**,**f**,**i**) Angular cross-correlation functions *C*(∆) measured at scattering vectors *s*_1_ (**c**), *s*_3_ (**f**), *s*_4_ (**i**) when frequency *v* = 2 is set to zero (red line) or when the two frequencies with *v* = 2 and *v* = 4 are set to zero (purple line) (adopted with permission from Mancini et al. [106] Copyright (2016) American Chemical Society).

**Figure 18 materials-12-03464-f018:**
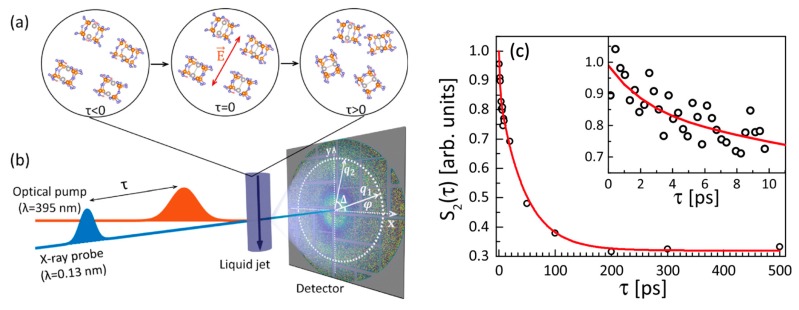
(**a**) Temporal evolution of an ensemble of randomly oriented tetrakis-μ-pyrophosphitodiplatinate (II) (PtPOP) molecules before (τ<0) and after (τ≥0) excitation. Only the molecules with the transition dipole moment (along the Pt-Pt axis) parallel to the laser field ***E*** are excited at τ=0; (**b**) Scheme of the optical pump X-ray probe experiment at LCLS. On the detector, the difference diffraction pattern averaged over N≈3000 shots is shown; (**c**) Temporal decay with the time constant 46±10 ps of the value $S_{2}\left(\tau \right)$, which is normalized area under the peak at $q=1.8\;{Å}^{-1}$ of the second Fourier component of the CCF $\langle C_{2}\left(q,\tau \right)\rangle$. The faster decay of $S_{2}\left(\tau \right)$ with the time constant 1.9±1.5 ps is shown in inset (adapted from [58]). Used under CC-BY 4.0.

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
