# Peer review of "Angular X-ray Cross-Correlation Analysis (AXCCA): Basic Concepts and Recent Applications to Soft Matter and Nanomaterials"

_materials, 2019, doi:10.3390/ma12213464_

Round 1

Reviewer 1 Report

This is a well-written review, well organized and with a very clear introduction.

The examples that have been chosen to describe the method and show what its potentiality is are very interesting and fully and comprehensively illustrate the technique.

I have no revision to request and in my opinion this manuscript can be published as it is. 

Author Response

We are thankful to the reviewer for her/his positive evaluation of the manuscript.

Reviewer 2 Report

The authors are doing an excellent job of summarizing this technique, from basic concepts to the applications. It is written well and the references include most of the significant reports in this area. I suggest accepting this review.

I was just wondering if the authors can give a summary of other techniques which can be used for this orientational order determination, and how they are different compared with AXCCA. 

Author Response

To address the question raised by the referee we added the following sentence in lines 306-308:

Angular anisotropies in x-ray or electron diffraction patterns may be analyzed as well by the multipole expansion of Legendre polynomials as suggested in the work  [53].

And we also added reference:

[53]    J. S. Baskin and A. H. Zewail, ChemPhysChem 7, 1562 (2006).

A detailed analysis of different approaches to study an angular anisotropy of diffraction patterns is out of scope of this review.

Reviewer 3 Report

Dear authors, 

The manuscript: “Angular x-ray cross-correlation analysis (AXCCA): Basic concepts and recent applications to soft matter and nanomaterials” The authors explained the angular x-ray analysis and its importance in soft matters using recent applications.

The manuscript is nicely written I will suggest author to look at some grammar part. Since this looks review report, author may need cite all figures with proper citation. 

Author Response

To reflect the review comment we now provided references to all Figures that were earlier published in other journals and completed it with the statements on publication rights.